# Multiscale Analysis of Cellular Composition and Morphology in Intact Cerebral Organoids

**DOI:** 10.3390/biology11091270

**Published:** 2022-08-26

**Authors:** Haihua Ma, Juan Chen, Zhiyu Deng, Tingting Sun, Qingming Luo, Hui Gong, Xiangning Li, Ben Long

**Affiliations:** 1Britton Chance Center for Biomedical Photonics, Wuhan National Laboratory for Optoelectronics, Huazhong University of Science and Technology, Wuhan 430074, China; 2Key Laboratory of Biomedical Engineering of Hainan Province, School of Biomedical Engineering, Hainan University, Haikou 570228, China; 3HUST-Suzhou Institute for Brainsmatics, Jiangsu Industrial Technology Research Institute, Suzhou 215123, China

**Keywords:** high-resolution imaging, fMOST, cerebral organoids, morphological analysis, spatial distribution

## Abstract

**Simple Summary:**

We have established a pipeline to analyze the structures of intact millimeter-scale cerebral organoids. By using this pipeline, the morphological and spatial distribution of neurons and GFAP-positive cells in organoids, as well as the spatial distribution of cortical neuron subtypes, were obtained by using fMOST imaging. This study introduced a new approach to monitor cellular composition and morphology of cerebral organoids.

**Abstract:**

Cerebral organoids recapitulate in vivo phenotypes and physiological functions of the brain and have great potential in studying brain development, modeling diseases, and conducting neural network research. It is essential to obtain whole-mount three-dimensional (3D) images of cerebral organoids at cellular levels to explore their characteristics and applications. Existing histological strategies sacrifice inherent spatial characteristics of organoids, and the strategy for volume imaging and 3D analysis of entire organoids is urgently needed. Here, we proposed a high-resolution imaging pipeline based on fluorescent labeling by viral transduction and 3D immunostaining with fluorescence micro-optical sectioning tomography (fMOST). We were able to image intact organoids using our pipeline, revealing cytoarchitecture information of organoids and the spatial localization of neurons and glial fibrillary acidic protein positive cells (GFAP^+^ cells). We performed single-cell reconstruction to analyze the morphology of neurons and GFAP^+^ cells. Localization and quantitative analysis of cortical layer markers revealed heterogeneity of organoids. This pipeline enabled acquisition of high-resolution spatial information of millimeter-scale organoids for analyzing their cell composition and morphology.

## 1. Introduction

Recently, many organoid model systems have emerged as the technology of human induced pluripotent stem cells (hiPSC) differentiation become widely available. These stem cells self-organize into 3D structures that resemble certain tissues. The scientific community has successfully cultured organoids that recapitulate the development of the brain [1,2], small intestine [3], kidney [4], retina [5,6], etc. Since organoids recapitulate the physiology and functional traits of in vivo tissues and organs, they are widely used in academic and clinical research fields, such as developmental biology, disease modeling, medicine testing, and personalized medicine [7]. Specifically, brain organoids have contributed significantly to research in developmental neuroscience and the study of neural diseases [8].

Brain organoids are important physiological models, and in order to understand the developmental and pathogenetic mechanisms behind organoids, it is essential to understand their cellular composition, cytoarchitecture, physiological processes, and cell–cell interactions. Traditionally, single-cell transcriptome analysis [9] and two-dimensional (2D) histology [10,11,12] have been the predominant methods for analyzing the characteristics of organoids. However, such methods could not reveal the spatial information of intact organoids. In addition, organoids are notoriously variable among individuals and batches [13]. Each organoid is slightly different as organoids exhibit independent regional development of “neuroepithelial units” [14], precluding the use of tissue atlas, on which 2D histology relies to locate structures. Thus, there is a pressing need for novel 3D whole-organoid imaging methods for experimental analyses.

A variety of 3D imaging protocols of organoids have been developed, many of which combined optical clearing methods with optical imaging techniques such as confocal microscopy and light-sheet microscopy. These optical clearing methods, such as BABB (a mixture of benzyl-alcohol and benzyl-benzoate) [11], SHIELD (stabilization under harsh conditions via intramolecular epoxide linkages to prevent degradation) [15], 2Eci (second-generation ethyl cinnamate-based clearing) [16], Scale (a chemical approach for fluorescence imaging and reconstruction of transparent mouse brain) [17], PACT (passive clarity technique) [18], and fructose–glycine [19], aim to increase imaging depth by matching the refractive index of the biological sample with immersion liquids, allowing 3D volume imaging. However, these methods require a variety of chemicals and multiple steps of rinses, often causing deformation of the organoids and weakening of fluorescent signals. In addition, over the years as the field of organoid research developed, organoids are becoming larger and more complex. The size of cerebral organoids could be several millimeters in diameter [20,21]. Time and efforts spent in sample processing is relative to the size of the organoids, and hence the processing of large cerebral organoids might require days or even weeks. It is challenging to perform high-resolution imaging of organoids larger than several millimeters in diameter using existing methods. There is a pressing need to develop a comprehensive 3D imaging protocol for large brain organoids that allows analysis of their characteristics and complexity.

Here, we provide a pipeline for labeling, embedding, imaging, and analyzing intact millimeter-scale cerebral organoids. We utilized fMOST, a 3D imaging approach based on semi-ultrathin sectioning tomography, to produce detailed 3D datasets of whole organoids. We achieved whole-volume 3D imaging of large cerebral organoids at single-cell resolution, reconstruction of single neurons and GFAP^+^ cells, and visualization of spatial localization of cortical layer markers. This method could become a valuable tool to reveal the cellular composition, structure, pathological changes, and even development of cerebral organoids and other organoids.

## 2. Materials and Methods

### 2.1. Culture of Cerebral Organoid

Human skin-derived iPSCs DYR0100 were purchased from Stem Cell Bank of Chinese Academy of Science. Organoids (Cat # HopCell-3D-C) were purchased from Hopstem Biotechnology Inc. (Hangzhou, China) and the generation of these organoids followed the published protocol [22]. All organoids were maintained in 5% CO_2_ incubators at 37 °C. They were cultured in 3D cerebral organoid medium (Cat # HopCell-3DM-100A) with 3D cerebral organoid medium supplement (Cat # HopCell-3DM-100B). Half of the medium was replaced every three days. For each group of viral transduction or 3D immunostaining experiments, we used more than five organoids. All organoids were harvested on day 110 for further research in this study. This study was approved by the Ethics Committee of Huazhong University of Science and Technology (Approval code: S613).

### 2.2. Lentiviral Transduction of Organoids

pLenti-hSyn-EGFP-P2A-MCS-WPRE were added into medium solution to label neurons, and pLenti-short GFAP-mCherry-WPRE or pLenti-GfaABC1D-EGFP-WPRE were used to label GFAP^+^ cells. The concentration of lentivirus mentioned above were 3 × 10^7^ TU/mL in general labeling, and the multiplicity of infection (MOI) was 10. When we performed sparse labeling of neurons, the lentivirus concentration was 1 × 10^7^ TU/mL. After 24 h of incubation after viral transduction, all the culture medium was changed. Seventy-two hours after viral transduction, the cultures were imaged using a confocal microscope to verify the expression of fluorescent proteins. All lentiviral tools were packaged by OBiO Technology Co., Ltd. (Shanghai, China).

### 2.3. 3D Immunostaining

We immunostained the intact cerebral organoids using iDISCO (immunolabeling-enabled three-dimensional imaging of solvent-cleared organs) protocols [23]. All samples were fixed in 4% paraformaldehyde solution for 2 h and then washed in phosphate buffer saline (PBS) for 1 h. A graded methanol/H_2_O series (20%, 40%, 60%, 80%, 100%, and a second 100%) were used to dehydrate samples for 30 min each at 4 °C. Samples were bleached in chilled 5% H_2_O_2_/20% DMSO/methanol overnight at 4 °C. Next, another graded methanol/H_2_O series (100%, 80%, 60%, 40%, and 20%) was used to rehydrate samples for 30 min each at 4 °C. Then, samples were incubated in PBS/0.5% Triton X-100 (PTX) for 12h at 4 °C. To perform immunostaining, samples were treated in PTX/20% DMSO/0.3 M glycine at 4 °C for 12 h, then blocked in PTX/10% DMSO/6% BSA at 4 °C for 12 h. Samples were washed in PBS/0.2% Tween-20 (PTw) at 37 °C overnight. Primary antibodies were diluted in PTw/5% DMSO/3% BSA, then samples were incubated with diluted primary antibodies for 4 days at 4 °C and washed 5 times for 1 h in PTw. Secondary antibodies were diluted in PTw/3% BSA, then samples were incubated with diluted secondary antibodies at 37 °C for 2 days, with attention paid to the evaporation of the liquid and the diluent secondary antibody replenished in time. The dilution ratios of primary and secondary antibodies are shown in Table 1. Finally, the samples were washed 5 times in PTw for 1 h.

### 2.4. Resin Embedding

The labeled organoids were embedded in Glycol methacrylate (GMA) resin [24]. First, samples were washed in PBS for 1 h five times, and then dehydrated in a graded ethanol series (50%, 75%, 95%, and a second 95%) for 30 min each at 4 °C. After dehydration, samples were immersed in a graded GMA series (50%, 70%, and 85%) for 1 h each at 4 °C, and impregnated in 100% GMA solution for 24 h at 4 °C. Finally, samples were transferred into gelatin capsules, and 100% GMA solution was added to fill the capsules. Then, the capsules were polymerized in a vacuum oven at 36 °C for 24 h.

### 2.5. Confocal Microscopy

To verify viral transduction and immunostaining, the labeled organoids were embedded in 5% agarose, sliced into 100 μm sections by vibration microtome, and sealed between two coverslips. Commercial confocal microscope (LSM710, ZEISS) was used to image the slices with 5× objective lens and 20× objective lens.

### 2.6. fMOST Imaging

The fluorescence micro-optical sectioning tomography (fMOST) [25] system was used to obtain dual-wavelength and high-resolution 3D data of organoids. During imaging, the sample was immersed in a water bath, and the solution was changed to propidium iodide (PI) solution during the collection of cytoarchitectonic information. The imaging system performed data acquisition at a voxel resolution of 0.32 × 0.32 × 2 μm^3^ with 20× objective lens. Mosaic-by-mosaic scans of the sample surface were performed by moving the 3D precision stage. After one layer of imaging was acquired, the imaged layer was excised. Then, the imaging system continued to acquire images of the next layer. The cycle repeats until the acquisition of complete organoid data was completed.

### 2.7. Visualization and Reconstruction

We converted microscopy images into tag image file format (TIFF) format and built z-stack using ImageJ software (NIH, Bethesda, MD, USA). We processed the data using Imaris (Bitplane, Zurich, Switzerland v9.0) to visualize 3D reconstruction. For data that requires quantitative statistics, we use ImageJ software to import the desired site cut into Imaris software to form a 3D data block for cell counting and reconstruction. We used the Filaments module to track cell processes to obtain 3D single-cell morphology, and then the parameters of single-cell can be obtained in the Statistics module, such as cell branch length, the number of branch points, and endpoints. Additionally, we used the Spots function to render cells in the organoid for cell localization analysis and number statistics.

## 3. Results

### 3.1. High-Resolution 3D Imaging Pipeline for Cerebral Organoids

To achieve high-resolution 3D imaging of cerebral organoids, we established a complete pipeline for labeling, embedding, imaging, and data analysis of cerebral organoids (Figure 1a). We labeled neurons and GFAP^+^ cells. GFAP^+^ cells in developing cerebral organoids may include astrocytes, radial glia cells, and neural precursor cells. We used two labeling strategies, viral transduction and 3D immunostaining, to demonstrate the distribution of neurons and GFAP^+^ cells in those organoids. The fluorescently labeled organoids were then embedded in GMA resin to be processed by fMOST to obtain serial semi-ultrathin sections. At last, we 3D reconstructed the dataset for analysis. We were able to obtain images of the structures of neuronal and GFAP^+^ cell processes in organoids and study their characteristics.

Both labeling strategies achieved labeling of target cells in organoids. Viral transduction enabled neurons and GFAP^+^ cells to express endogenous enhanced green fluorescent protein (EGFP) and mCherry (a basic red fluorescent protein), respectively. We were able to identify processes of GFAP^+^ cells and neurite outgrowths as long as 100 μm (Figure 1b). The second strategy was the immunostaining method in iDISCO [23]. The entire organoids were immunostained with fluorescent antibodies that label neurons, GFAP^+^ cells, and cortical layers. The processes of GFAP^+^ cells were also clearly visible, and we could observe the processes in the center of the organoids (Figure 1c). Both labeling strategies were compatible with the pipeline. 

Images from confocal microscopes have a high x- and y-axis resolution, but it was difficult to acquire 3D data of GFAP^+^ cells in intact organoids using confocal microscopes since the imaging depth was only around 200 μm (Figure 1d). By utilizing the fMOST system, we obtained a continuous dataset with a spatial resolution of 0.32 × 0.32 × 2.00 μm^3^, allowing 3D reconstruction of entire organoids. In the 3D view of the organoid, we were able to visualize the spatial distribution of GFAP^+^ cells in intact organoids (Figure 1e). We managed to perform 3D imaging and analyses of cell composition in organoids using the pipeline presented. 

### 3.2. Visualization of Fine Structures of Neurons and GFAP^+^ Cells

We reconstructed the continuous fMOST dataset of cerebral organoids for presentation and cell morphology analysis. We were able to observe neurons with EGFP expression at various distances from the surface of the organoid based on serial sections (Figure 2a). Both the somas (Figure 2(b1,b2)) and long processes (Figure 2(b3,b4)) of neurons were well-defined. After visualizing the spatial distribution of neurons (Figure 2c), we estimated the number of EGFP^+^ neurons to be 1.42 × 10^4^ using spot analysis in Imaris software (Figure 2d). Morphological analysis of GFAP^+^ cells revealed that several branched processes extended from the somas of GFAP^+^ cells to different directions (Figure 2e). In our raw imaging data with higher magnification, cell bodies and fibers were clearly visible (Figure 2(e1–e3)). A 3D view of GFAP^+^ cells illustrated an extensive network of GFAP^+^ cells with long processes (Figure 2f). The complex network structures of neurons and GFAP^+^ cells effected a compelling representation of the intricate cellular system in the brain. We were able to visualize the complex cellular networks of organoids and analyze cell morphology using our high-resolution 3D dataset. 

### 3.3. Propidium Iodide (PI) Staining Provided Cytoarchitecture Information 

Cytoarchitecture information acts as macro-anatomical landmarks and enables precise localization for other fluorescent markers. In this study, we used PI, nucleic acid dye, to illustrate cell density and positioning of cell nuclei in organoids. The organoids were immersed in PI solution during the imaging process using fMOST to obtain cytoarchitectural information. PI-stained images showed that nuclei were unevenly distributed in organoids, indicating that the distribution of cells in organoids was not uniform (Figure 3a). For instance, in selected areas, fewer nuclei were located along the border than away from the border (Figure 3b). Nuclei differed in sizes in different areas (e.g., smaller in Figure 3b and larger in Figure 3c). 

To reveal any potential distribution patterns of nuclei across multiple organoids, we embedded and imaged arrays of organoids, which also significantly decreased processing time. In images of these arrays, we observed that each organoid was different (Figure 3d). We identified rosette-like structures in z-projections of 60-μm-thick sections (Figure 3d,e). These signature structures often indicate the presence of neural stem cells. Researchers usually immunostained their cerebral organoids with PAX6 or SOX2 to further demonstrate the rosette structures [2]. In addition, PI staining offered information on the number of cells in organoids and enabled spatial localization for signals in other channels (Figure 3f). In 3D reconstructed images, we observed irregular aggregation of GFAP^+^ cells on the surface of organoids (Figure 3g). We were able to obtain cytoarchitecture information of multiple organoids in an array and use the information to analyze the distribution of other markers.

### 3.4. Single-Cell Reconstruction Using 3D Datasets of Organoids

We reconstructed single neurons (Figure 4a) and GFAP^+^ cells (Figure 4d) in organoids to analyze their morphology. Single-cell morphology of neurons was uncovered by lentivirus labeling followed by cell tracking using Imaris software (Figure 4b,c). In the images we acquired, we could easily identify GFAP^+^ cells with 3D immunostaining (Figure 4d,e) and locate two GFAP^+^ cells that spanned more than 420 μm (Figure 4f). In order to analyze the complex morphological structures of the two traced GFAP^+^ cells, we calculated their degrees of branching (Figure 4g,h), lengths of each branch (Figure 4i), and the number of branch points and terminal points (Figure 4j). We analyzed these data and learned that the two cells had different morphological characteristics. Comparing with GFAP^+^ cell 2, GFAP^+^ cell 1 had more primary processes, a higher number of end points, and a simpler branching structure. Both GFAP^+^ cells have processes over 300 μm (Figure 4g,h). We were able to perform single-cell reconstruction in organoids and analyze the complexity of branched processes of GFAP^+^ cells. 

### 3.5. Spatial Localization of Cortical Layer Markers in Organoids

We employed immunostaining to analyze the spatial distribution of differentiated cortical neurons in organoids. First, we imaged sliced organoids using confocal microscopes and observed the presence of neurons expressing superficial layer marker special AT-rich sequence binding protein 2 (SATB2) and neurons expressing deep layer marker T-box brain protein 1 (TBR1) in cerebral organoids (Figure 5a). Immunostained sliced organoids showed that higher amount of SATB2 and TBR1 were expressed in the outer layer than in the center of the organoids. The cerebral organoids displayed layered structures like the cerebral cortex [26]. However, organoid slices could not provide us a full view of the distribution of differentiated neurons in the entire organoid. Thus, we performed 3D imaging and reconstruction to obtain spatial distribution of SATB2^+^ and TBR1^+^ cells. We noticed that their distribution was not uniform across the organoid, with some cells expressing TBR1 intensively while others expressed SATB2 (Figure 5b).

Next, we analyzed the spatial localization patterns by spot analysis (Figure 5c). In this sampled volume, the density of labeled cells increased toward the periphery of the organoid, and SATB2^+^ cells were almost absent near the center (Figure 5d). We then verified this distribution pattern by graphing SATB2^+^ and TBR1^+^ cells in an organoid (Figure 5e), corresponding to the distribution patterns shown in immunostained sliced organoids. We quantified the spatial distribution of labeled cells by graphing the numbers of cells along the radius of the organoid. More than 95% of labeled cells were located between 400 and 500 μm away from the center of the organoid (Figure 5f). No SATB2^+^ cells were located within 100 μm from the center point, but TBR1^+^ cells were present all the way along the radius, which corresponded to the development of cortex. We were able to observe cortical layering and analyze the spatial localization of differentiated cortical neurons in organoids using our pipeline.

## 4. Discussion

In this report, we presented research framework for fluorescent labeling, sample preparation, and high-resolution 3D imaging of intact cerebral cortical organoids, which allowed analyses of sub-cellular structures as well as cell composition of organoids. In designing our protocol, we optimized two fully compatible fluorescent labeling strategies, viral transduction and 3D immunostaining. Single-cell reconstruction utilizing high-resolution data offered fine details of cell morphology. We were able to obtain information on spatial localization of differentiated neurons and GFAP^+^ cells in relation to cytoarchitecture in organoids. Importantly, our protocol enabled comprehensive in situ study on large organoids with diameters of several millimeters [11]. Our method enabled visualization and quantification of spatial features of intact organoids, providing valuable tools for studying organoid models of fetal brain development [2,27], biocompatibility of materials [26,28], and neurotoxicity of medicine [29] as well as basic research on cell morphology [30] and neuronal connections [31].

We primarily used the fMOST technology to image organoids in 3D. Cerebral organoids are valuable tools in studying neuronal projections as they could be used to recreate the connections between different brain regions in co-cultures of different brain region-specific organoids [31,32,33]. Along with current trends, organoid systems are becoming larger and hence more difficult to image. Although optical imaging techniques combined with tissue clearing has been widely applied in imaging organoids, these techniques do have certain limitations, such as sample shrinkage, poor preservation of fluorescent signals, and low efficiency with large intricate organoids in some cases [17]. Confocal microscopy and two-photon microscopy have been used to reconstruct the morphology of single cells with the advantage of being able to detect neuronal activities in real time for organoids, but as the size of organoids increases, it is challenging to image entire organoids [34]. Therefore, we chose to use fMOST to obtain 3D images of intact organoids. As a physical sectioning imaging method, it does not require tissue clearing, which makes it ideal for imaging larger-volume organoids.

Our workflow using fMOST also allowed single-cell reconstruction and had potential to benefit a variety of studies focusing on cell morphology. Tracking the changes of neuronal morphology in organoid systems modeling microcephaly and Alzheimer’s disease helped researchers study pathological progresses [35,36]. However, most studies utilized 2D histology on sliced organoids. Our workflow allowed 3D single-cell reconstruction with a higher level of details in single neurons and GFAP^+^ cells, enabling visualization of changes in cell morphology at different stages of differentiation in diseases. Organoids are known to be inherently varied both among and within batches [37], so it is important to quickly process a large number of organoids in quantitative and comparative studies. We were able to obtain images of multiple organoids embedded in the same resin block, which significantly increased the throughput of the protocol (as shown in Figure 3f). However, it was difficult to arrange organoids in specific patterns for imaging. We are developing a plastic mold for easy and accurate placement of multiple organoids during embedding and have obtained some preliminary results using prototypes.

We labeled organoids using two methods, lentiviral transduction and 3D immunostaining. Viral transduction is only possible if there is a suitable virus with a promotor corresponding to the specific cell types. If not, we had to turn to immunostaining. Organoids are still alive after lentivirus labeling, allowing continuous real-time observation of single cell development. There is a much larger set of available antibodies than viruses, so immunostaining allows more types of markers as well as their co-labeling. Notably, in developing cerebral organoids, GFAP is not only a marker for astrocytes, but also possibly for radial glia cells and some neural precursor cells. In addition, PI staining provided cytoarchitectural information and cellular localization. A rosette-like structure was observed in our data (Figure 3d), and interestingly, the two cortical layer markers SATB2 and TBR1 did not exhibit such structure. In the future, we will perform co-immunostaining for stem cell markers (Sox2 or Pax6) and cortical layer markers to confirm these rosette structures, and the spatial distribution of cortical layer markers will help us analyze these rosette structures.

Our labeling methods could still be improved. When transducing the organoids, the lentiviruses labeled too many neurons, disabling us from easily distinguishing single cells for reconstruction. We reduced the amount of lentiviruses added in an effort to achieve sparse labeling, but fewer lentiviruses resulted in incomplete labeling of neurites (Figure 4c). Potentially, we could use adeno-associated viruses (AAVs) for sparse labeling to identify individual neurons or astrocytes and their long-range projections [38]. In addition, stereotactic injection in organoids might help label neurons in specific areas of organoids, facilitating analysis of heterogeneity of organoids. Both schemes may be used to label single neurons with abundant neurites in organoids. As for 3D immunostaining, it took at least ten days to immunostain intact cerebral organoids. It would be beneficial to fine-tune sample processing parameters, such as temperature and types of solvents, to shorten the processing time. That way, we can deal with larger and more complex organoids in the future. 

In summary, we were able to image intact cerebral organoids, obtain details on cell composition and morphology in organoids, reconstruct single neurons and GFAP^+^ cells, and determine the distribution of specific markers. This pipeline could be easily adapted to be used on other organoids besides cerebral organoids. We are still working on improving the pipeline, hoping to contribute more to the organoid research community, which facilitates further studies on cell morphological changes, fiber projections, and cell-cell interactions within organoids. This protocol could be used to study organoid disease models, accelerating towards clinical treatment of neurodegenerative diseases, cancers, and other diseases.

## 5. Conclusions

We presented a pipeline for studying the cellular composition and morphological structure of intact organoids that combines viral transduction, 3D immunostaining, and fMOST imaging. Using this method, we obtained morphological structures of neurons and GFAP^+^ cells and were able to observe rosette-like structures in organoids. This study introduced a practical method for cerebral organoid imaging and associated quantification. This pipeline has the potential to be applied to cytoarchitectural studies of other types of organoids.

## Figures and Tables

**Figure 1 biology-11-01270-f001:**
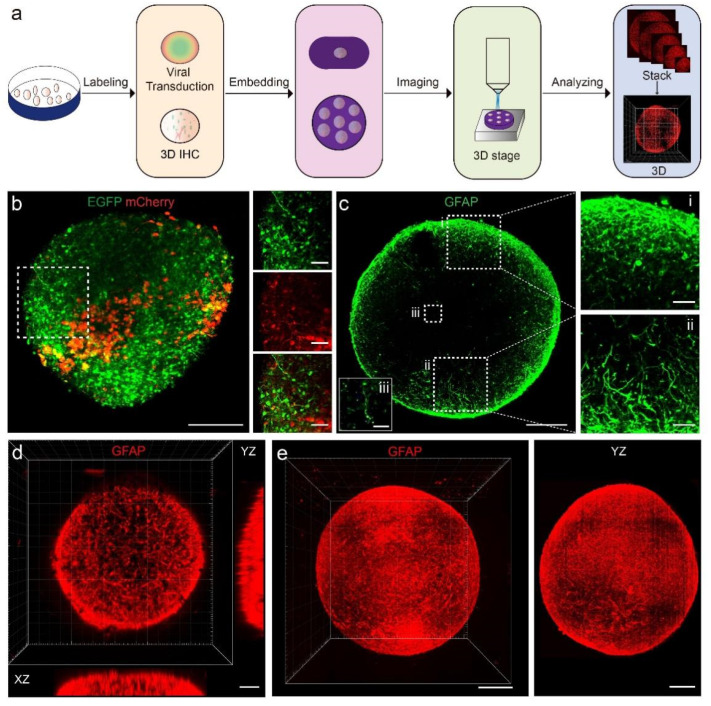
Pipeline for 3D imaging of cerebral organoids. (**a**) Steps for labeling, embedding, imaging, and analyzing of organoids. (**b**) Maximum intensity z-projections of 100-μm-thick sections showing expression of EGFP and mCherry in neurons and GFAP^+^ cells after viral transduction. Scale bars: 200 μm (left) and 50 μm (right). (**c**) Maximum intensity z-projections of 100-μm-thick sections showing GFAP^+^ cell after 3D immunostaining. Scale bars: 200 μm (**c**); 50 μm (**i**,**ii**); 25 μm (**iii**). (**d**,**e**) 3D view of GFAP^+^ cell in an organoid. (**d**) was obtained by confocal microscopy, (**e**) was obtained by fMOST. Scale bar: 200 μm.

**Figure 2 biology-11-01270-f002:**
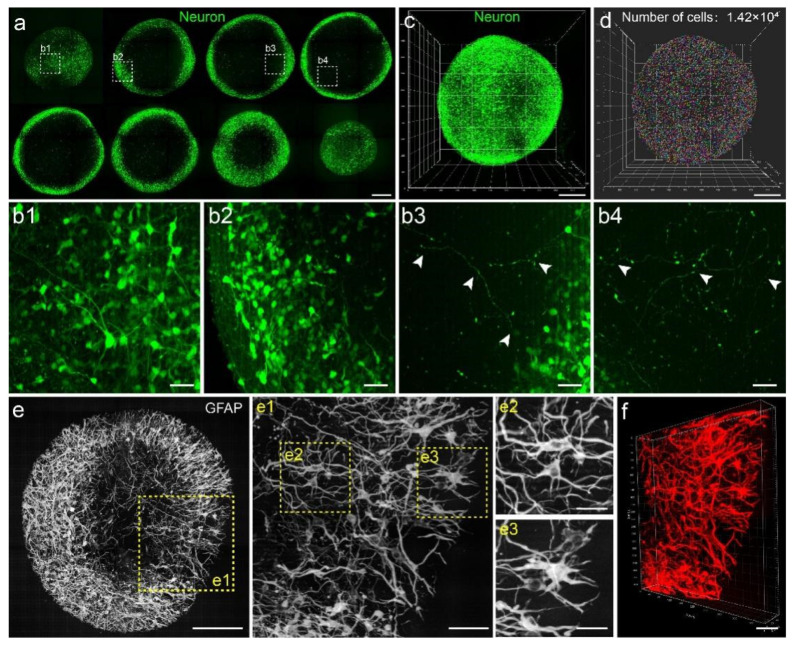
Single-cell morphology analysis in cerebral organoids. (**a**) Maximum intensity z-projections of 200-μm-thick sections from an organoid showing neurons labeled with EGFP. Scale bar: 200 μm. (**b1**–**b4**) Higher magnifications of specified regions in (**a**). (**b1**,**b2**) showed the somas, and the arrows in (**b3**,**b4**) indicated the neuronal processes. Scale bars: 50 μm. (**c**) Three-dimensional view of neurons in an organoid. Scale bar: 200 μm. (**d**) Spot analysis showing the number of neuronal somas. Scale bar: 200 μm. (**e**) Maximum intensity z-projection of a 60-μm-thick section of an organoid showing GFAP^+^ cells. (**e1**–**e3**) were higher magnifications of specified regions in (**e**). Scale bars: 200 μm (**e**); 50 μm (**e1**); 25 μm (**e2**,**e3**). (**f**) Three-dimensional view of the area showed in (**e1**). Scale bar: 200 μm.

**Figure 3 biology-11-01270-f003:**
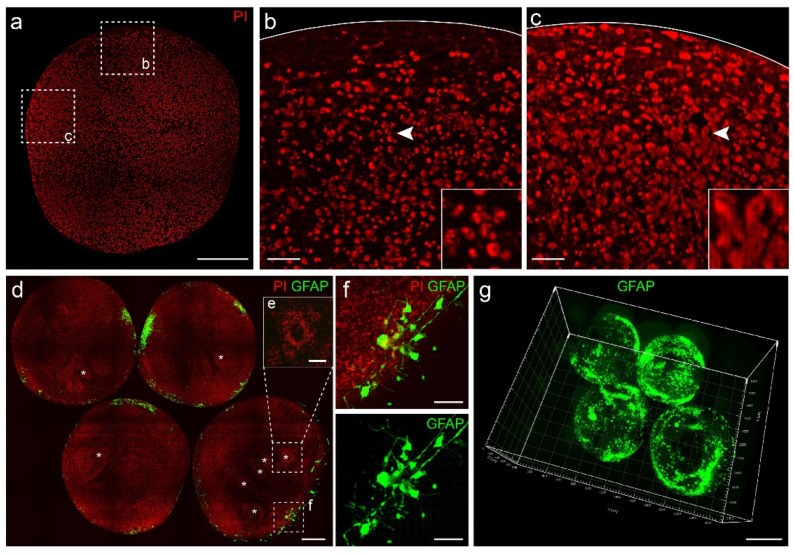
PI staining results and comparison of multiple organoids. (**a**) Maximum intensity z-projection of a 2-μm-thick section of a PI-stained organoid. Scale bar: 200 μm. (**b**,**c**) Higher magnifications of specified regions in a. Arrows indicated the location of enlarged views. Scale bars: 25 μm. (**d**) Standard deviation z-projection of a 60-μm-thick section of an array of organoids. Asterisks indicated rosette-like structures (**e**). Scale bar: 200 μm. (**f**) Higher magnifications of the specified region in d. Scale bar: 50 μm. (**g**) 3D reconstruction of GFAP^+^ cells in the array of multiple organoids from (**d**). Scale bar: 500 μm.

**Figure 4 biology-11-01270-f004:**
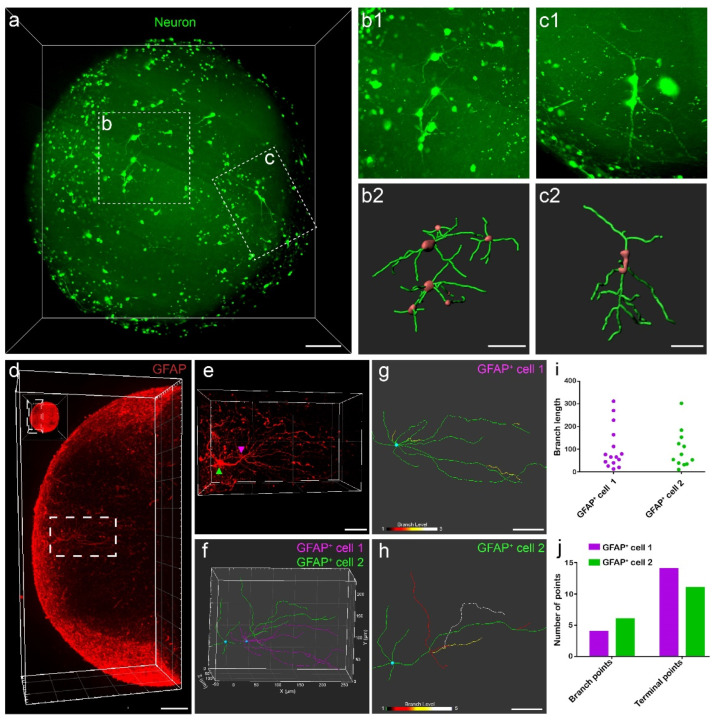
Three-dimensional imaging dataset of organoids enables single-cell reconstruction. (**a**) Three-dimensional view of neurons in cerebral organoids. Scale bar: 100 μm. (**b1**,**c1**) Higher magnifications of specified regions in (**a**). (**b2**,**c2**) Neuron morphology reconstructed with Imaris software. Scale bar: 50 μm. (**d**) Three-dimensional view of GFAP^+^ cells in organoids. Scale bar: 100 μm. (**e**) Higher magnifications of specified regions in d. Scale bar: 50 μm. (**f**) Single-cell reconstruction of two GFAP^+^ cells. (**g**,**h**) Branch level of each astrocyte. Scale bar: 50 μm. (**i**) A scatter plot of their branch lengths of the two tracked GFAP^+^ cells. (**j**) Histogram showing the numbers of branch points and terminal points of the two tracked GFAP^+^ cells. Green fluorescent signals in all graphs in this figure denoted neurons, and the red fluorescent signals denoted GFAP^+^ cells.

**Figure 5 biology-11-01270-f005:**
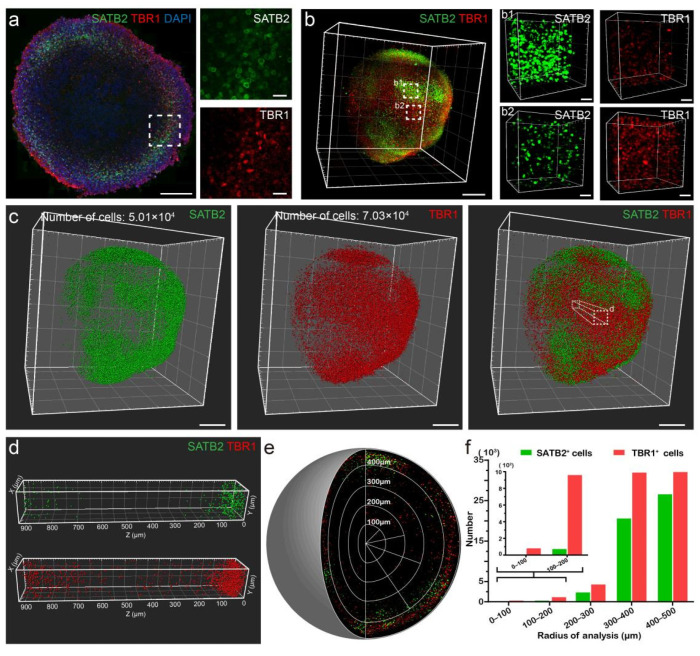
Distribution of cortical layer markers in organoids. (**a**) Maximum intensity z-projections of 100-μm-thick sections of a cerebral organoid imaged by confocal microscopy. Scale bar: 200 μm (left); 25 μm (right). (**b**) Three-dimensional view shows the spatial distribution of SATB2^+^ and TBR1^+^ cells in a whole organoid. (**b1**,**b2**) Higher magnifications of specified regions in b. Scale bars: 200 μm (left); 50 μm (right). (**c**) Spot analysis showing the number of SATB2^+^ and TBR1^+^ cells. Scale bars: 200 μm. (**d**) Sample volume from (**c**) showing details on the distribution of SATB2^+^ and TBR1^+^ cells along the radius of the organoid. (**e**) Distribution of SATB2^+^ and TBR1^+^ cells within the organoid. (**f**) Graph showing the spatial distribution of SATB2^+^ and TBR1^+^ cells, illustrating increased cell density toward the periphery of organoids. In this figure, the red fluorescent signals denoted TBR1^+^ cells, and the green fluorescent signals denoted SATB2^+^ cells.

**Table 1 biology-11-01270-t001:** Antibodies used in the study.

Antibody	Dilution	Cat. No.	Company
Anti-SATB2, mouse	1:200	ab51502	Abcam, Cambridge, UK
Anti-TBR1, rabbit	1:200	ab31940	Abcam, Cambridge, UK
Anti-GFAP, rabbit	1:200	Z0334	Dako, Glostrup, Denmark
Anti-NeuN, rabbit	1:200	ab177487	Abcam, Cambridge, UK
Alexa Fluor 488 donkey anti-mouse lgG (H + L)	1:500	A21202	Invitrogen, Waltham, MA, USA
Alexa Fluor 594 donkey anti-rabbit lgG (H + L)	1:500	A21207	Invitrogen, Waltham, MA, USA
Alexa Fluor 488 donkey anti-rabbit lgG (H + L)	1:500	A21206	Invitrogen, Waltham, MA, USA

## Data Availability

All data related to this study are contained within the manuscript. Data can be obtained from the corresponding authors on request.

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
