# Peer review of "Multiscale Analysis of Cellular Composition and Morphology in Intact Cerebral Organoids"

_biology, 2022, doi:10.3390/biology11091270_

Round 1
Reviewer 1 Report
Ma et.al described a very useful method for labeling, embedding, imaging, and analyzing intact millimeter-scale cerebral organoids. By using fMOST, they got the larger cerebral organoids 3D imaging at single-cell resolution, deconstructed the spatial localization of neurons and astrocytes, and provided a potential tool for exploring kinds of brain diseases. The study is well designed and the manuscript is well written. There are some concerns authors should address to improve the current manuscript.
Minor questions:
1. The authors mentioned the pipeline they described could become valuable tools for investigating the developmental and pathological changes of cerebral organoids in the ending of introduction. It would be better to give an example for this suppose. The cerebral organoids could model many development and pathological changes, they just need to verify their method could work in one of brain developmental or pathological events, which would be a big plus for this work.
2. Many abbreviations were not standard: line 49-50, line 76, line 80, line 87, line 148, line 301, line 339
3. The English should be improved.
Author Response
Sincerely thanks for your comments! The response point-by-point are as follows:
Question 1: The authors mentioned the pipeline they described could become valuable tools for investigating the developmental and pathological changes of cerebral organoids in the ending of introduction. It would be better to give an example for this suppose. The cerebral organoids could model many development and pathological changes, they just need to verify their method could work in one of brain developmental or pathological events, which would be a big plus for this work.
Response: Thanks for your thoughtful suggestion. In this study, we focused on establishing strategy to analyze the structures of intact millimeter-scale cerebral organoids. We also indeed performed LPS neuroinflammation experiments on the organoids. However, we did not observe apparent morphological changes at single-cell level after LPS treatment, and this part of the data was not shown in the manuscript. We will refine the experiments of the organoid development or pathological models in future research.
Question 2: Many abbreviations were not standard: line 49-50, line 76, line 80, line 87, line 148, line 301, line 339.
Response: We have added the full name of the abbreviations and check this question in the full revised manuscript (as shown in lines 58-63, 106-107, 170-171, etc).
Question 3: The English should be improved.
Response: Following your suggestion, we invited colleagues who are proficient in English to help revise the manuscript and made corresponding revisions.
Reviewer 2 Report
Interesting methodological work, which can indicate a protocol for the microscopic analysis of organoids, basically based on two labeling strategies, viral transduction and 3D immunostaining.
Summary
Adequate and easy to follow that invites the reader to read the article
Introduction
Clear and easy to follow and read correct for the manuscript
Results
That being from a methodological approach, they could even be described in more detail, incubation times used in viral infections, amount of virus, concentrations and times of antibodies, be deeper in all the methodological development
In point 2.1 the author refers to ultrathin sections, in microscopy this refers to sections of nanometers not microns, the scale used corresponds to semi ultrathin sections.
The author speaks of analyzing the fine structure of the organoids, to comment and be true at this point the author should at least analyze the morphology of the dendrites and thus we could approach the analysis of the fine structure by confocal microscopy.
Point 2.2 the authors speak of fine structure, I believe that this statement would be correct if we were talking about images at higher magnification (60x or 100x) that would allow us to see in detail the fine structure of the cells that make up the organoid.
The discussion is adequate, but it should be remembered that a tool for the analysis of organoids by confocal microscopy is presented.
Author Response
Sincerely thanks for your comments! The response point-by-point are as follows:
Question 1: That being from a methodological approach, they could even be described in more detail, incubation times used in viral infections, amount of virus, concentrations and times of antibodies, be deeper in all the methodological development.
Response: Thanks for your advice. We describe the detail protocols and experimental parameters of viral infections and immunostaining of organoids which can be found in “Lentiviral transduction of organoids” and “3D immunostaining” section (lines 95-104; lines 105-122 and Table 1).
In brief, the Lenti-viruses incubation time in this study was 24h, and the MOI was 10 for conventional infection (lines 98-101). Antibodies concentrations were shown in Table 1, and the incubation time of primary antibodies and secondary antibodies were 4 days and 2 days, respectively (lines 116-121).
Question 2: In point 2.1 the author refers to ultrathin sections, in microscopy this refers to sections of nanometers not microns, the scale used corresponds to semi ultrathin sections.
Response: Thanks for your comment. The thickness of the slices during fMOST imaging was 1-2 μm in this study, we have revised it in the manuscript to semi-ultrathin sections (lines 76-78, 164-166).
Question 3: The author speaks of analyzing the fine structure of the organoids, to comment and be true at this point the author should at least analyze the morphology of the dendrites and thus we could approach the analysis of the fine structure by confocal microscopy.
Point 2.2 the authors speak of fine structure, I believe that this statement would be correct if we were talking about images at higher magnification (60x or 100x) that would allow us to see in detail the fine structure of the cells that make up the organoid.
The discussion is adequate, but it should be remembered that a tool for the analysis of organoids by confocal microscopy is presented.
Response: Thanks for your comments. In this manuscript, we called subcellular structures like neuronal dendrites and fibers of GFAP+ cells as “fine structures”, and we also reconstructed and analyzed such structures as shown in Section 3.4 “Single-cell reconstruction using 3D datasets of organoids” and Figure 4. When performing fMOST imaging, we generally use a 20× objective lens. In fact, the dataset of organoids in this study often showed the complete single-cell morphology with high-resolution, such as in Figure 1 and Figure 2. We don’t need higher magnification to detect cell morphology or spatial distribution. To avoid understanding bias, we modified “fine structure” in the revised manuscript.
We are aware that confocal microscopy could be used to analyze organoids with high resolution in the x-y plane, but there is limited z-axis imaging depth, which makes the technology unsuitable for millimeter-scale organoids analysis (lines 178-180). Therefore, we proposed this imaging pipeline to enable the analysis of large-volume organoids. We have added the discussion on confocal or two-photon microscopy in lines 326-329.
Reviewer 3 Report
In the present study, Ma et al., applied state-of-art fMOST technique to examine the morphological characteristics and spatial distribution of selective cell subtypes in three-dimensional cerebral organoids. The study is well designed and the experiments are well conducted. The authors reported a series of interesting findings including the 3-D morphological reconstruction and analysis of GFAP-expressing cells as well as the spatial distribution of cortical neuron subtypes. However, I have major concern regarding the interpretation of GFAP positive cells as astrocytes throughout the manuscript. A wealth of literatures reported the expression of GFAP in neural stem cells during cortical development (Kriegstein and Alvarez-Buylla. Annu Rev Neurosci. 2009;32:149-84.). Therefore, GFAP expression was not sufficient to demonstrate the identity of astrocytes in organoids. GFAP may also label radial glial cells in developing cerebral organoids. The authors should perform co-immunostaining with stem cell markers Sox2 and Pax6, as pointed out in the maintext, to distinguish radial glial cells from astrocytes, or revise the astrocytes in the maintext as "GFAP-positive cells".
The following few minor comments should also be addressed:
1. In Figure 3d, zoom-in images or representative markers for rosette should be provided to demonstrate the presence of rosette structure in organoids.
2. In Figure 5, the authors should analyze the spatial distribution of cortical layer markers around each rosette, the equivalent ventricular zone of the developing cerebral cortex in cerebral organoids.
3. On Page 6, line 182, "deep marker" should be revised as "deep layer marker".
Author Response
In the present study, Ma et al., applied state-of-art fMOST technique to examine the morphological characteristics and spatial distribution of selective cell subtypes in three-dimensional cerebral organoids. The study is well designed and the experiments are well conducted. The authors reported a series of interesting findings including the 3-D morphological reconstruction and analysis of GFAP-expressing cells as well as the spatial distribution of cortical neuron subtypes. However, I have major concern regarding the interpretation of GFAP positive cells as astrocytes throughout the manuscript. A wealth of literatures reported the expression of GFAP in neural stem cells during cortical development (Kriegstein and Alvarez-Buylla. Annu Rev Neurosci. 2009;32:149-84.). Therefore, GFAP expression was not sufficient to demonstrate the identity of astrocytes in organoids. GFAP may also label radial glial cells in developing cerebral organoids. The authors should perform co-immunostaining with stem cell markers Sox2 and Pax6, as pointed out in the maintext, to distinguish radial glial cells from astrocytes, or revise the astrocytes in the maintext as "GFAP-positive cells".
Response: Thanks for your sincere comments and suggestion. We have read this paper carefully and realized that neurons and glial cells might be derived from the same precursor cells, which also express the GFAP protein. In addition, some developing astrocytes and some subpopulations of astrocytes in adults with some degree of multipotency may be directed to differentiate into neurons. Therefore, we cannot define GFAP-positive cells as astrocytes.
To be more precise, we have revised “astrocytes” in the manuscript as “GFAP+ cells” and explained the cell types that GFAP+ cells might comprise (lines 161-162, 352-354). For further validation of astrocytes, we will perform more experiments in future studies.
Question 1: In Figure 3d, zoom-in images or representative markers for rosette should be provided to demonstrate the presence of rosette structure in organoids.
Response: Thanks for your suggestion. We have modified the Figure 3 and added enlarged view of the rosettes structure of organoids in Figure 3d so that our images can be more intuitive (line 228).
Question 2: In Figure 5, the authors should analyze the spatial distribution of cortical layer markers around each rosette, the equivalent ventricular zone of the developing cerebral cortex in cerebral organoids.
Response: Thanks for your suggestion. In this study, we used cytoarchitecture information by PI staining to demonstrate rosette structure. However, cortical layer markers occupied imaging channels and we did not obtain the PI cytoarchitecture information in Figure 5. We can’t well identify the rosettes structure in this experiment, and in our data, these cortical layer markers were not evident around the rosettes. We added the discussion on this question (lines 355-359).
Question 3: On Page 6, line 182, "deep marker" should be revised as "deep layer marker".
Response: Sorry for this mistake. We have revised as “superficial layer marker” or “deep layer marker” in manuscript (lines 273-274).
Round 2
Reviewer 2 Report
endorsed publication of this manuscript